# Enhancing Cutin Extraction Efficiency from Industrially Derived Tomato Processing Residues by High-Pressure Homogenization

**DOI:** 10.3390/foods13091415

**Published:** 2024-05-04

**Authors:** Elham Eslami, Francesco Donsì, Giovanna Ferrari, Gianpiero Pataro

**Affiliations:** 1Department of Industrial Engineering, University of Salerno, Via Giovanni Paolo II, 132-84084 Fisciano, Italy; eeslami@unisa.it (E.E.); fdonsi@unisa.it (F.D.); gferrari@unisa.it (G.F.); 2ProdAl Scarl, University of Salerno, Via Giovanni Paolo II, 132-84084 Fisciano, Italy

**Keywords:** tomato processing residues, alkaline hydrolysis, high-pressure homogenization (HPH), cutin, response surface methodology, thermal properties, SEM

## Abstract

This study primarily aimed to enhance the extraction of cutin from industrial tomato peel residues. Initially, the conventional extraction process was optimized using response surface methodology (RSM). Subsequently, high-pressure homogenization (HPH) was introduced to improve extraction efficiency and sustainability. The optimization process focused on determining the optimal conditions for conventional extraction via chemical hydrolysis, including temperature (100–130 °C), time (15–120 min), and NaOH concentration (1–3%). The optimized conditions, determined as 130 °C, 120 min, and 3% NaOH solution, yielded a maximum cutin extraction of 32.5%. Furthermore, the results indicated that applying HPH pre-treatment to tomato peels before alkaline hydrolysis significantly increased the cutin extraction yield, reaching 46.1%. This represents an approximately 42% increase compared to the conventional process. Importantly, HPH pre-treatment enabled cutin extraction under milder conditions using a 2% NaOH solution, reducing NaOH usage by 33%, while still achieving a substantial cutin yield of 45.6%. FT-IR analysis confirmed that cutin obtained via both conventional and HPH-assisted extraction exhibited similar chemical structures, indicating that the main chemical groups and structure of cutin remained unaltered by HPH treatment. Furthermore, cutin extracts from both conventional and HPH-assisted extraction demonstrated thermal stability up to approximately 200 °C, with less than 5% weight loss according to TGA analysis. These findings underscore the potential of HPH technology to significantly enhance cutin extraction yield from tomato peel residues while utilizing milder chemical hydrolysis conditions, thereby promoting a more sustainable and efficient cutin extraction process.

## 1. Introduction

Cutin, constituting 40–85% of the plant cuticle by weight, acts as an outer protective layer for leaves, aerial parts, and fruits [1]. It features an amorphous branching structure, forming a flexible three-dimensional polymer network of C_16_ to C_18_ fatty acids linked by ester bonds [2]. This network interacts with polysaccharides (primarily cellulose and pectin), waxes, phenolic compounds, and aromatic components [2,3]. Functionally, cutin serves as a structural component, defending against pathogens [4] and preventing water loss alongside waxes [5]. It also aids substance transport across plant tissues [6]. Cutin possesses unique characteristics such as being non-toxic, biodegradable, waterproof, UV-blocking, amorphous, insoluble, and infusible, making it a valuable bio-polyester. These distinct properties have spurred research interest in synthetic compounds mimicking cutin-based polymers for applications in packaging, UV filters, and membranes [7].

However, many plant-based biomasses offer abundant and cost-effective sources of natural cutin. If efficiently extracted, this resource could provide novel building blocks with specific reactive polyfunctional characteristics, making them suitable for pharmaceutical applications [2]. Furthermore, they could be utilized in sustainable material development, such as innovative bio-resins and lacquers serving as internal coatings for metal food cans as an alternative to bisphenol A (BPA) resins [8].

Tomato pomace, a byproduct of tomato processing, consists of skins, seeds, and pulp [9,10]. It typically makes up 2–5% of the total weight of processed tomatoes [9,11], with about 33% seeds, 27% skin, and 40% pulp [12,13,14]. Globally, it amounts to around 1.2 million tons annually, with Europe alone generating approximately 0.3 million tons [3]. Disposal of these residues poses challenges, with current low-value uses including animal feed, fertilizer substrate, and biogas production [9,11]. Some extraction for lycopene, a potent antioxidant, also occurs for use in food, pharmaceutical, and cosmetic products [11].

However, tomato pomace, especially the peels, also serves as a major reservoir of cutin [2,14], as well as fiber [15,16], and carbohydrates such as cellulose and pectin [16,17]. Remarkably, tomato pomace contains about 20% cutin by weight, with an estimated annual cutin volume ranging from 10^5^ to 10^6^ tons [3]. This underscores the significant potential for extracting cutin from tomato peels, which could significantly impact the development of sustainable industrial practices.

Efforts have recently focused on extracting cutin from tomato processing waste to create renewable products [1,2,14,18]. The conventional method for extracting cutin typically involves depolymerization, breaking it down into individual monomers rather than extracting it whole. One common depolymerization method, firstly patented by Cigognini et al. [19] for extracting cutin from discarded tomato peels, involves the direct hydrolysis of the plant material. Specifically, it employs a hot alkaline solution of sodium hydroxide to break down the cutin polymer embedded in tomato peel followed by precipitation of cutin monomers through an acidification process. This extraction method, currently considered the industry standard [8,20], has been effectively utilized by various researchers for extracting cutin from tomato processing residues, particularly for manufacturing bio-resins, hydrophobic edible films [2,18,21], and bio-based polymers [14]. Nevertheless, a literature gap persists, as previous studies have only provided a broad exploration of experimental conditions regarding temperature (ranging from 65 to 130 °C), time (spanning from 15 min to 24 h), and concentration (ranging from 1 to 4%) of alkaline (NaOH) solution for the hydrolysis process involved in cutin extraction [2,14], without addressing their optimization.

Moreover, while alkaline hydrolysis has proven effective for cutin depolymerization, there is a growing need for milder processing conditions (e.g., reduced alkali and acid concentrations). In this scenario, various innovative methods, such as pulsed electric fields (PEFs), high-pressure homogenization (HPH), supercritical fluid extraction (SFE), and ultrasound-assisted extraction (US) have emerged as promising alternative or complementary techniques for enhancing the extraction and retrieval of valuable compounds from diverse agri-food residues [9,22].

Among these techniques, HPH has received significant attention due to its capacity to rapidly and effectively micronize plant tissue in suspension, thereby unlocking bioactive compounds entrapped within cells with high extraction yields. The intense fluid-mechanical stresses (shear, elongation, turbulence, and cavitation) exerted on plant biomass suspended in water when passing through a specifically designed homogenization valve have been shown to significantly reduce particle size with a homogeneous size distribution, increasing the surface-to-volume ratio and enhancing the disruption of cellular structures, thus facilitating the extraction and release of valuable compounds [16,23]. In particular, HPH technology has demonstrated effectiveness in valorizing various agri-food wastes, including tomato processing residues, for the recovery of proteins and bioactives from tomato peels [24], β-carotene from tomato puree [25], and cellulose and pectin from tomato pomace [16,17]. However, no investigation has been reported so far into the application of HPH technology to improve the efficiency of cutin extraction from tomato peels or other fruit- and vegetable-based materials. Additionally, according to available literature, only one study has investigated the use of innovative techniques such as ultrasound-assisted extraction (UAE) for recovering cutin from different bio-wastes, including apple, tomato peel, and watermelon peel [26].

This study aimed to address the identified knowledge gaps by establishing the optimal conditions for conventional cutin extraction from industrial tomato peel residues. Response surface methodology (RSM) was employed to optimize the conventional cutin extraction process from tomato peels by investigating the influence of time, temperature, and NaOH concentration during the alkaline hydrolysis step on the recovery yield of cutin. Furthermore, the study aimed to explore the potential of combining HPH technology with a milder chemical hydrolysis process to develop an efficient and sustainable method for isolating cutin from tomato peels. The isolated cutin samples underwent comprehensive characterization using structural analysis techniques, such as Fourier transform infrared spectroscopy (FT-IR), as well as thermal analysis methods, including differential scanning calorimetry (DSC) and thermogravimetric analysis (TGA). Additionally, the impact of HPH on the microstructural surface of the tomato peels and extracted cutin samples was investigated through scanning electron microscopy (SEM).

## 2. Materials and Methods

### 2.1. Raw Material and Sample Preparation

Tomato pomace, consisting of approximately 39.4% peels, 10.6% seeds, and 50.0% pulp on a wet basis, was provided by a local factory located in the province of Salerno during the 2023 processing season. A sample of about 20 kg of wet tomato pomace was collected at the discharge point of the tomato pulper machine utilized in the tomato puree processing line. The sample was promptly transported to the laboratories of ProdAl Scarl (Fisciano, Italy) and stored under refrigeration (at T = 4 °C) until further use. Upon arrival at the laboratory, the moisture content of the tomato pomace was determined to be 83.0 ± 1.4%.

For the cutin extraction experiments, tomato peels were separated from the pomace using a homemade flotation-cum-sedimentation system. The system consisted of one mixing tank equipped with an impeller with three paddles for mixing pomace and water in a ratio of 1:30 (kg/L), along with two settling tanks. The separation process, lasting approximately one hour under ambient conditions, efficiently separated the tomato peels from the seeds and pulp. The seeds settled at the bottom of the container, while fibrous residues accumulated in the middle. In contrast, the peels floated on the surface due to their lower density. Subsequently, the floating peels were collected and passed through a trommel to remove excess water before being stored for cutin extraction. The moisture content on the tomato peel samples was determined to be 80 ± 2% on a wet basis.

### 2.2. Chemicals

Sodium hydroxide (NaOH) used for cutin depolymerization and hydrochloric acid (37% HCl fuming) used for the isolation and purification of the cutin monomers were purchased from PanReac (Barcelona, Spain).

### 2.3. Conventional Cutin Isolation Procedure

Cutin was isolated from tomato peels following the conventional extraction procedure outlined by Cigognini et al. [19], with minor adjustments, as illustrated in Figure 1.

Initially, approximately 50 g of tomato peels was placed in a 200 mL glass bottle containing an alkaline (NaOH) solution of a specified concentration, as detailed in the subsequent experimental design section. The solid-to-liquid ratio was maintained at 1:4 g/mL, as per the methodology outlined in previously published work [18]. To promote solvent–solid interaction, the mixture underwent high shear mixing (HSM) at 20,000 rpm for 10 min using a T-25 Ultra Turrax device (IKA^®^-Werke GmbH & Co. KG, Staufen, Germany) equipped with an S25-N18 G rotor in an ice bath to prevent temperature increases. Subsequently, thermal treatment was conducted in an autoclave device (ALFA-10-Plus, PBI, Italy, Milan) at a specified temperature and duration, as described in the experimental design section (2.4).

After alkaline hydrolysis, the exhausted peels were removed, and the remaining dark brown solution underwent filtration using a 25 µm metal mesh filter to eliminate the residual biomass and suspended solids. The filtered liquid was then acidified by adding 37% (*v*/*v*) HCl fuming until reaching a final pH of 4–4.1 at 20 °C. This pH adjustment precipitated cutin monomers, resulting in a color change from brown to ochre. The acidified solution was subsequently centrifuged at 5289× *g* (PK130R model, ALC International, Cologno Monzese, Italy) for 20 min at 10 °C to separate the liquid phase from the gummy brown residues containing cutin monomers. This operation was repeated three times by re-dispersing the monomeric solid mixture in distilled water to remove residual acid. This procedure resulted in brown gummy paste, with a moisture content of approximately 65 ± 5% on a wet basis.

### 2.4. Experimental Design for Optimization of Cutin Extraction by Conventional Method

In this study, response surface methodology (RSM) was employed to establish the relationship between the response variable (cutin yield) and the main independent factors involved in the alkaline hydrolysis step, as well as to determine the optimal conditions for maximizing the extraction yield of cutin from tomato peels.

To achieve this objective, a three-factor face-centered central composite design (FCCCD) with six center points was constructed. This design aimed to investigate the effects of the concentration of the alkaline (NaOH) solution (ranging from 1% to 3%, *w*/*v*), along with the time (ranging from 15 to 120 min) and temperature (ranging from 100 to 130 °C) of the thermal treatment phase, on the extraction yield of cutin from tomato peels. The upper and lower limits of the temperature, time, and NaOH concentration domains were chosen based on the literature findings [2,14,19] to cover a significant range of values. The extraction yield was evaluated gravimetrically using Equation (1).
(1)$$Yield\left(\%\right)=\frac{M_{c}}{M_{TP}}\cdot 100$$
where *M_C_* and *M_TP_* are, respectively, the weight of the extracted cutin and the initial tomato peel sample (based on the dry weight).

In total, the experimental design encompassed 14 distinct runs, as detailed in Table 1.

A second-order polynomial model was used to predict the response variable as a function of the three independent factors, as described by Equation (2).
(2)$$Y=\beta_{0}+\sum_{i=1}^{3}\beta_{i}X_{i}+\sum_{i=1}^{2}\sum_{j=i+1}^{3}\beta_{ij}X_{i}X_{J}$$
where Y is the predicted response variable; X_i_ and X_j_ are the independent variables; and β_0_, β_i_, and β_ij_ are the intercept, the regression coefficient of the linear term, and the regression coefficient of the interaction term of the model, respectively.

### 2.5. HPH-Assisted Extraction of Cutin

The implementation of the HPH pre-treatment of tomato peels aimed to enhance the recovery yield of cutin during the conventional extraction process under the optimal thermal processing conditions previously determined through RSM analysis.

As illustrated in Figure 1, prior to undergoing HPH treatment, approximately 50 g of tomato peels was suspended in distilled water (at a ratio of 1:11 g/mL) in a 1000 mL Erlenmeyer flask and subjected to HSM at 20,000 rpm for 10 min in an ice bath to prevent any temperature increases. Subsequently, the tomato peel suspension underwent HPH using an in-house developed system, as detailed elsewhere [24]. The suspension, with an inlet temperature of 20 ± 2 °C, was forced to pass through an orifice valve assembly with an orifice diameter of 200 μm at a pressure of 80 MPa for 10 passes. This condition for HPH treatment is based on the research of Pirozzi et al. which demonstrated its effectiveness in improving cellulose extraction from tomato pomace [16]. To maintain the product temperature below 25 °C after each pass, a tube-in-tube heat exchanger was installed immediately upstream and downstream of the orifice valve.

Following HPH treatment, the suspension was subjected to vacuum drying at 40 °C using a Büchi R300 Rotavapor (BUCHI Italia s.r.l., Cornaredo, Italy) to remove water and achieve a paste with a final humidity of approximately 78 ± 2%, closely matching the moisture content of wet tomato peels (80 ± 2%) directly subjected to the conventional extraction process. The concentrated HPH-treated tomato peel samples (50 g) then underwent cutin isolation through the optimized conventional extraction process, described in the previous section. Additionally, to explore the potential of HPH as an alternative or complementary approach to chemical hydrolysis, the cutin isolation procedure following HPH pre-treatment was conducted at different NaOH concentrations (ranging from 1% to 3%, *w*/*v*) under the optimized conditions of temperature and hydrolysis time established for the conventional process.

### 2.6. Fourier Transform Infrared Spectroscopy (FT-IR)

FT-IR spectroscopy was used to acquire the characteristic spectra of the cutin samples isolated in both conventional and HPH-assisted extraction processes using an FT-IR-4100 series spectrophotometer (Jasco Europe S.r.l., Cremella, Italy) equipped with a single-reflectance horizontal attenuated total reflectance (ATR) cell (ATR-PRO 470-H, Jasco Europe S.r.l., Cremella, Italy). The infrared spectra were recorded in the 4000–650 cm^−1^ range at a resolution of 4 cm^−1^ after 16 scans. The resulting spectrum was based on at least three repetitions, which were averaged and smoothed utilizing the adaptive-smoothing function.

### 2.7. Thermal Characterization

The thermal properties of cutin samples obtained through both conventional and HPH-assisted extraction from tomato peels were evaluated using both differential scanning calorimetry (DSC) and thermogravimetric analysis (TGA), following the procedures reported by Cifarelli et al. [18] with slight modifications.

In DSC analysis, measurements were conducted to determine the glass transition temperature (T_g_), crystallization temperature (T_c_), and melting temperature (T_m_) of the samples using a DSC822e instrument (Mettler Toledo, GmbH, Greifensee, Switzerland), calibrated with an indium standard. The measurement began by subjecting 3–5 mg of each sample to a cooling phase from 25 °C to −60 °C, followed by a holding time of 2 min at −60 °C. Subsequently, the sample was heated to 150 °C and then cooled again to −60 °C. All heating/cooling phases were carried out at a rate of 50 °C/min under a nitrogen gas (N_2_) flow rate of 50 mL/min.

TGA analysis was performed using a TC-10 thermobalance (Mettler Toledo, GmbH, Greifensee, Switzerland). For each measurement, approximately 5 mg of each sample was heated from 25 °C to 700 °C at a heating rate of 10 °C/min under an air atmosphere. Several parameters, including the maximum decomposition temperature (T_dmax_), percentage of solid residue at 700 °C, and derivative thermograms (DTG), were evaluated. DSC and TGA analyses of the cutin obtained by different extraction procedures were carried out in triplicate.

### 2.8. Morphological Analysis

The impact of HSM and HPH pre-treatments on the morphological characteristics of tomato peels and cutin extracts was assessed through scanning electron microscopy (SEM), as described by Pirozzi et al. [16]. Briefly, the samples were mounted to an aluminum stub and coated with a 10 nm thick layer of gold–palladium alloy via sputter coating before examination under a high-resolution ZEISS HD15 Scanning Electron Microscope (Zeiss, Oberkochen, Germany) at 1000× magnification.

### 2.9. Statistical Analysis

All the experiments and analyses were conducted in triplicate to calculate the mean and standard deviation (SD) for each set of experimental data. Differences with statistical significance between the means were determined using one-way ANOVA and Tukey’s Honest Significant Difference test, where a p-value of less than 0.05 was considered significant. These analyses were carried out with SPSS statistical software (version 20, SPSS Inc., Chicago, IL, USA). Furthermore, the factorial central composite design (FC-CCD) and subsequent data evaluation were executed using Design Expert software, Version 12 (Minneapolis, MN, USA).

## 3. Results

### 3.1. Conventional Extraction of Cutin from Tomato Peels

Table 1 shows the effect of the combinations of three independent variables derived from the FC-CCD, namely temperature (100–130 °C), time (15–120 min), and NaOH concentration (1–3%), on the recovery yield of cutin from industrial tomato peel residue during the conventional extraction process.

The obtained results indicate that all three independent factors significantly influenced the investigated response variable within the examined range. Notably, the concentration of the alkaline (NaOH) solution emerged as the principal factor affecting the extraction yield of cutin, especially noticeable with concentrations above 1%. The highest cutin yield (32.5%) was detected when the most intense conditions (130 °C, 120 min, 3% NaOH) were applied. Conversely, employing the same thermal treatment conditions (130 °C, 120 min) with the lowest NaOH concentration (1%) resulted in a markedly lower cutin yield of only 5.6%.

These findings align with prior studies underscoring the pivotal role of NaOH concentration in cutin depolymerization. Previous studies indicate that hydroxide ions (OH^−^) from NaOH target the carbonyl carbon within the ester bond, and NaOH effectively compromises the structural integrity of cutin, facilitating its extraction process [27,28]. Furthermore, it is noteworthy that the effect of NaOH concentration intensified with increasing temperature or duration of thermal treatment, indicating a positive interaction among these variables. For instance, at a fixed duration of 120 min, elevating NaOH concentration from 1% to 3% led to a significant rise in cutin extraction yield, from 5.8% to 23.5% at 100 °C and from 5.6% to 32.5% at 130 °C. Similarly, at a constant temperature of 130 °C, increasing NaOH concentration from 1% to 3% resulted in an increase in cutin extraction yield from 3.9% to 21.1% after 15 min and from 5.6% to 32.5% after 120 min.

The results also demonstrate that the impact of thermal treatment conditions (time and temperature) significantly influenced cutin yield only when using NaOH concentrations greater than 1%. Similarly, Benítez et al. [14] found that the yield percentages of the cutin extract depended on both hydrolysis time and temperature. For instance, after 24 h of hydrolysis, the values increased from 25 to 32% at 70 and 100 °C, respectively.

Additionally, the findings presented in Table 1 highlight a compensatory effect between the time and temperature variables during the thermal treatment. Specifically, it was observed that, at a constant NaOH concentration, higher temperatures combined with shorter durations could produce results comparable to those achieved with lower temperatures over longer periods of heating. For instance, maintaining a consistent NaOH concentration of 3%, thermal treatment at 100 °C for 120 min resulted in a yield of approximately 23.5%, which was not statistically different from the 23.4% yield obtained at 115 °C for 67.5 min. This finding aligns with observations reported by Cifarelli et al. [2] who noted that when maintaining a constant NaOH concentration of 3%, the resulting process yield achieved at 100 °C for 6 h (20%) did not show a significant difference compared to that obtained (18%) at 130 °C for 15 min.

The data obtained from the FC-CCD were fitted to a second-order polynomial equation (Equation (2)). Since some of the terms of the equation were not statistically significant, these terms were excluded from the general evaluation of the model, thus reducing the overall complexity of the mathematical relationship between independent and response variables. The values and significance of the regression coefficients of the predicted polynomial models and corresponding *p* values, determination coefficient (R^2^), adjusted R^2^, and Root Mean Square Error (RMSE) are reported in Table 2.

The results show that all the investigated factors exerted a significant linear effect on the cutin extraction yield, with the concentration of NaOH and hydrolysis time demonstrating the most substantial impact on the response (*p* ≤ 0.001). Furthermore, positive statistically significant interactions were observed between NaOH concentration and both time and temperature (*p* ≤ 0.05), affecting the cutin yield. This provides evidence that the combined effect of these variables is indeed greater than what would be expected from their individual effects alone, thus supporting the presence of a synergistic effect. In contrast, it was found that temperature exhibited a non-significant (*p* > 0.05) dependence on the time. Therefore, this term was not taken into consideration for the general evaluation of the model, thus simplifying the mathematical correlation between independent and response variables.

The ANOVA results (Table 2) show that the RSME value was 0.00268 and that the determination coefficient (R^2^) and adjusted R^2^ value for the relationship between the response variable and the alkaline hydrolysis parameters were 0.9578 and 0.9217, respectively. These values signify a strong correlation between observed and predicted data. Additionally, the analysis of variance demonstrated that the model used was significant (*p* ≤ 0.0002) for the response variable and the lack of fit test was not significant (*p* > 0.05), supporting the predictive efficacy of the selected model.

The three-dimensional response surface plots depicted in Figure 2 show the interaction of the hydrolysis time and temperature and NaOH concentration on the recovery yield of cutin from tomato peels upon conventional extraction process, which aligns well with previous findings [2,14]. In particular, in accordance with the coefficients and significance of each factor involved in the model, the graphs clearly demonstrate that cutin yield increased almost linearly with an increase in the independent factors. Furthermore, even though all the investigated variables had a statistically significant effect on cutin yield (*p* ≤ 0.05), NaOH concentration was the factor that mostly influenced the observed response, in agreement with the higher value of the linear coefficient of the NaOH concentration than that of temperature and time (Table 2). Moreover, the influence of thermal treatment conditions, and especially that of hydrolysis time, on the cutin extraction appeared more pronounced only at NaOH concentrations higher than 1%. Beyond this threshold value, there is noticeable evidence of a synergistic effect among the alkaline hydrolysis parameters. These results may be ascribed to the capability of NaOH to cleave the ester bonds, thus facilitating cutin extraction [27,28].

Moreover, the obtained results allowed the definition of the optimal conditions for the conventional extraction process within the investigated domain resulting in the highest extraction yield of cutin. Specifically, the values of the factors maximizing the response variable were determined by the adopted model to be 130 °C, 120 min, and 3% NaOH, resulting in a cutin yield of 32.5%. However, further experiments should involve increasing the upper limits of the input variables to ascertain whether these extreme processing conditions truly represent the optimum ones.

While comparing extraction yields of cutin from tomato processing residues with those previously reported in the literature is challenging due to various factors, such as tomato variety, type of tomato residue, equipment, and experimental protocols, the results obtained in this study are consistent with the current literature. For example, Cifarelli et al. [2] found that alkaline hydrolysis of tomato peels at 130 °C for 2 h using a 3% NaOH solution resulted in the highest cutin yield of 28%, which was 1.4 and 1.6 times higher than that achieved under thermal treatment conditions of 100 °C for 6 h and 130 °C for 15 min, respectively.

### 3.2. Effect of HPH-Assisted Extraction on the Recovery Yield of Cutin

The integration of HPH pre-treatment of tomato peels before conventional cutin extraction, conducted under optimized conditions, aimed to enhance the cutin recovery while evaluating the interaction between the HPH process and NaOH concentration, which resulted in a more relevant factor than time and temperature. This assessment sought to explore the feasibility of combining HPH pre-treatment with a milder chemical hydrolysis approach to achieve the desired extraction yield.

Tomato peels subjected to HSM were further treated with HPH before conventional cutin extraction through alkaline hydrolysis, maintaining optimal thermal conditions (130 °C for 120 min) with varying NaOH concentrations (1%, 2%, and 3%).

The results in Table 3 demonstrated that, regardless of NaOH concentration, HPH pre-treatment significantly (*p* < 0.05) increased cutin yield compared to conventional extraction. Interestingly, combining HPH pre-treatment with 1% NaOH concentration led to a significant increase in cutin yield up to 33.5%, six-fold higher than that achieved with conventional extraction under the same conditions. Moreover, at NaOH concentrations of 2% and 3%, HPH-treated samples exhibited cutin yields of 45.6% and 46.1%, respectively, 1.8- and 1.4-fold higher than corresponding conventional extraction yields. This integration of HPH technology either enabled significantly higher cutin yields under optimized extraction conditions or allowed for a concurrent reduction in NaOH concentration during alkaline hydrolysis, while maintaining the same extraction yield as conventional extraction, and in the need for HCl during the monomer precipitation phase. Notably, no significant difference (*p* > 0.05) was observed in cutin yields from HPH-treated samples when NaOH concentration varied from 2% to 3%. Therefore, 2% NaOH concentration was determined as optimal, maximizing cutin yield to 40.3%. Additionally, it reduced the usage of NaOH (by 33%) during alkaline hydrolysis and HCl (by 35%) during the subsequent monomer precipitation step compared to the optimized conventional extraction process.

The highest cutin content observed in HPH-treated samples confirmed the contribution of HPH to improving the cutin isolation process by depolymerizing cutin in an alkaline solution. This improvement is attributed to HPH’s ability to micronize plant tissue in suspension [24,29], creating favorable conditions for subsequent chemical hydrolysis and enhancing extraction efficiency, even at low NaOH concentrations.

This finding aligns with previous results, which demonstrated that under processing conditions comparable to those utilized in this study, HPH treatment (80 MPa, 10 passes) reduced particle sizes of tomato pomace to a range of 55–90 µm, while HSM treatment resulted in a broader size range of 580–1020 µm. Furthermore, the variation in particle size induced by HPH pre-treatment also enhanced suspension stability [29].

While no prior work has specifically addressed HPH-assisted extraction of cutin from agri-food wastes, previous studies have demonstrated HPH’s efficacy in enhancing extraction yields by reducing particle size and promoting solvent interaction in various applications, including sulforaphane extraction from broccoli seeds [30], cellulose extraction from tomato peels [16], and pectin recovery from tomatoes [17]. For example, Pirozzi et al. [16] demonstrated that HPH pre-treatment of tomato pomace, with its ability to micronize plant tissue in suspension, created favorable conditions for subsequent chemical hydrolysis and facilitated high-yield recovery of cellulose. Additionally, they observed that HPH contributed to enhancing cellulose yield by loosening fibril aggregation, breaking bonds between lamellae, and promoting defibrillation. Similarly, Van Audenhove et al. [17] investigated the utilization of HPH-assisted acid extraction to recover pectin from tomatoes. Their findings indicated significant alterations in the polysaccharide cell walls due to HPH treatment (at 20 MPa for a single pass) resulting in the recovery of nearly two-thirds of the remaining pectin during the subsequent extraction phase. These observations suggest that HPH may facilitate the deconstruction of the tomato peel cuticular layer, which consists of a complex network of cutin, wax, and carbohydrates, enhancing cutin recovery from the fragmented cuticle layer.

The significant (*p* < 0.05) difference between conventional and HPH-assisted extraction reported in Table 3 was visually confirmed by photos of tomato peels, as shown in Figure 3, illustrating the visual appearance of tomato peels before and after HSM treatment and after HPH treatment, as well as that of the non-hydrolyzable residue left after alkaline hydrolysis.

The results unequivocally demonstrate that both mechanical methods, namely HSM alone and the combination of HSM-HPH, resulted in smaller particle sizes compared to non-micronized samples. However, the HSM-HPH-treated tomato peels exhibited a more uniform appearance compared to those treated solely with HSM. This phenomenon is ascribed to HPH’s capability to produce small particle sizes and uniform size distribution within agri-food residue-in-water suspensions [24]. This uniformity likely contributed to the enhanced interaction of small particles of tomato peels with the alkaline solution during the hydrolysis process, resulting in a decrease in the amount of the non-hydrolyzable part of tomato peels (residue obtained after the hydrolysis process and separated through filtration). Consequently, the cutin extraction from HSM-HPH-treated tomato peels surpassed that from HSM-treated ones.

This assertion was validated by quantifying the non-hydrolyzable fraction of tomato peels treated with HSM (Figure 3C) and HSM-HPH (Figure 3E), which revealed that 38% of the total HSM-treated tomato peel remained non-hydrolyzable, while only 16% of the total HSM-HPH-treated tomato peel was non-hydrolyzable. This suggests that a larger portion of the tomato peel was converted into the hydrolyzable part, thereby augmenting the extraction of cutin.

Overall, these results highlight the potential of HPH pre-treatment to enhance the extractability of cutin from tomato peels while employing gentler chemical hydrolysis conditions, thereby promoting a more sustainable and efficient cutin extraction process. However, further studies are necessary to optimize the processing conditions of this innovative extraction approach in order to fully exploit the benefits of HPH pre-treatment.

### 3.3. FT-IR Spectra to Define the Chemical Structure of Cutin

FT-IR spectroscopy serves as a valuable tool for analyzing chemical structures by identifying the functional groups of various materials and assessing structural modifications resulting from applied treatments.

Figure 4 shows the FT-IR spectra of cutin achieved through conventional (Figure 4—blue line) and HPH-assisted extraction processes (Figure 4—red line) under the optimized conditions (130 °C, 120 min, 3% NaOH). The identification of the most relevant signals in the FT-IR spectra of cutin extracted by conventional and HPH-assisted extraction methods is detailed in Table 4. The FT-IR spectra of extracts from both untreated and HPH-treated tomato peels exhibited common bands, including hydroxyl, methylene, and ester groups, which are typical of the cutin matrix [2,14,21,26,31], with slight differences between the two samples. In particular, the presence of cutin was confirmed by the prominent characteristic band at 1705 cm^−1^ (for untreated samples) and 1704 cm^−1^ (for HPH-treated samples) associated with the carbonyl stretching of carboxylic acid ν(C=O). Furthermore, shoulders at 1730 cm^−1^ and 1712 cm^−1^ were related to normal and conjugated ester carbonyls, respectively, while additional bands at around 1176 cm^−1^ and 1104 cm^−1^ indicated asymmetrical (νa) and symmetrical (νs) v(C-O-C) stretching ester vibrations, respectively, suggesting the presence of the cutin matrix, linking the different hydroxy fatty acids to create the cutin cross-linking [18]. The wide band observed at around 3300 cm^−1^ corresponded to the stretching vibration of the hydroxyl group [18], while the bands within the 3000–2800 cm^−1^ range were attributed to asymmetrical and symmetrical stretching vibrations of methylene (CH_2_) group at 2930 cm^−1^ and 2854 and 2851 cm^−1^, respectively, accompanied with δ(CH_2_) at 1463 cm^−1^ and δ(CH_2_) at 723 and 724 cm^−1^. These findings collectively suggest the existence of long-chain fatty acids in extracted cutin [26,31] from both conventional and HPH-assisted methods. In the infrared spectrum, absorption bands around 1605–1635 cm^−1^ and 900–800 cm^−1^ were attributed to the functional groups or structural properties of phenolics and flavonoids, likely contained in the cutin extract [18,26], with ν(C=C) stretching vibrations of phenolic acids at 1632 and 1635 cm^−1^, ν(C-C) aromatic at 1605 and 1606 cm^−1^, and (C-H) aromatic at 835 cm^−1^ [18]. The FT-IR analysis revealed that cutin extracts achieved in both conventional and HPH-assisted extraction processes presented a specific esterification degree and consisted of a combination of acids and esters, indicating that the chemical groups and cutin structure were not altered or destroyed by the mechanical treatment. More specifically, the FT-IR analysis revealed that the cutin samples from HPH-assisted extraction predominantly consisted of long-chain hydroxy acids and certain polysaccharides originating from non-hydrolyzable cutin. However, transmittances in cutin extracted by conventional extraction methods were sharper, especially for the 1400–1650 cm^−1^ region related to aromatic and C=C functional groups from phenolic and flavonoid compounds [14], suggesting that HPH treatment affects the chemical bonds of these compounds and weakens their structure.

### 3.4. Thermal Characterization of Cutin

Cutin, as a natural polymer, holds significant relevance in polymer science and material engineering. This study explored the thermal properties of cutin extracts derived from both untreated and HPH-treated tomato peels using thermogravimetric analysis (TGA) and differential scanning calorimetry (DSC). The primary aim was to investigate the thermal stability, key thermal degradation stages, and phase transitions of the cutin samples, which are crucial for cutin’s utilization in diverse applications, including innovative composite materials [32].

TGA provides insights into thermal stability and compositional alterations by tracking the weight loss against temperature variations, while differential thermogravimetry (DTG) offers details on the rates and precise temperatures of these changes. In the presented thermogravimetric profiles (Figure 5), both untreated and HPH-treated cutin samples exhibited similar multiple stages of weight loss, typically around five, consistent with the characteristic behavior often observed during surface evaporation for surface biomass-derived samples [2]. The initial weight loss observed (below 100 °C) in the DTG profiles of both samples was attributed to the presence of water [18]. Subsequent losses (around 120–150 °C) suggested water adsorption by hydroxyl cutin groups, which functioned as sites for coordinating water molecules. This observation was supported by the presence of hydrogen-bonded hydroxyl bands confirmed in the FT-IR analysis (Table 4). This unique characteristic of cutin may be explained by the plasticizing effect of the absorbed water [18]. However, the remaining peaks in the DTG curve did not directly correlate with the humidity levels of the samples. Both cutin samples from conventional and HPH-assisted extraction from tomato peels exhibited thermal stability up to approximately 200 °C, with less than 5% weight loss according to the TGA graph. This weight loss corresponds to the release of physical and chemical water within the cutin samples [33]. The peaks observed between 200 and 320 °C corresponded to processes such as decarboxylation and dehydration/esterification involving hydroxylated fatty acids [14,18], as well as cellulose in the sample, which was related to non-hydrolyzable cutin composition [2]. Subsequent secondary decomposition stages (between 360 °C and 475 °C) occur in multiple steps, constituting approximately 90% of the overall weight loss. This behavior is typical of complex biomass-derived samples composed of diverse mixtures of hydroxy acids [2]. The final stage of decomposition, similar to the stepwise decomposition pattern observed in estolides derived from acid oils, may be attributed to the fragmentation of the aliphatic chain of fatty acids with varying long chains or their oligomers [2,14,34].

Comparative analysis with conventional plastics revealed that extracted cutin exhibited a relatively high initial degradation temperature (200 °C) comparable to that of rubber and polyethylene (PE) and surpassing that of other plastics like poly (vinyl alcohol) (PVA), poly (vinyl chloride) (PVC), and polystyrene (PS) [27,33]. This noteworthy thermal stability is attributed to cutin’s inherent three-dimensional and heavily cross-linked structure [27]. This structural characteristic was previously demonstrated in FTIR analysis, which revealed interactions through hydrogen bonding of the ester group within the cutin polymer matrix, suggesting its potential in developing films or coatings with remarkable barrier or hydrophobic properties. In summary, understanding the thermal behavior of cutin extracts enhances their application potential in various fields, offering opportunities for innovative biomaterials and contributing to sustainable material development strategies.

The DSC results for cutin extracts from untreated (conventional process) and HPH-treated tomato peels are summarized in Table 5.

In DSC analysis, the glass transition temperature (T_g_) serves as a crucial parameter, indicating the temperature at which a material transitions from a rigid, glassy state to a more flexible, rubbery state [35]. This transition is associated with significant changes in the physical and mechanical properties of the material. In this study, T_g_ assessments were exclusively conducted on dried samples to prevent interference caused by freezable water, which can affect the observed T_g_ in cutin thermograms [18]. The T_g_ values, determined during the initial cycle of cooling/heating, revealed that the T_g_ value for cutin obtained from the untreated sample is approximately −30.2 °C, while the T_g_ value for cutin derived from HPH-treated tomato peels was observed around −19.7 °C. Both findings align with previous research indicating that the T_g_ of cutin extracted from tomato peel is below −20 °C [14,18,36]. The difference between the T_g_ values of untreated and HPH-treated samples can be attributed to the coextraction of cellulose with cutin due to the HPH treatment of tomato peels. The rigid, crystalline structure of cellulose could potentially enhance the overall structural integrity and stiffness of the cutin matrix, leading to changes in T_g_.

In the second heating/cooling cycle, the crystallization temperature (T_c_) was determined to be 1.5 °C for cutin extracted from HPH-treated tomato peels. In contrast, cutin extracts derived through conventional extraction methods did not exhibit a crystallization point. This phenomenon could be explained by the simultaneous extraction of cellulose, known for its crystalline structure [37,38], during HPH-assisted extraction leading to a modification in the originally wholly amorphous configuration of cutin [39]. Additionally, the increased melting point (T_m_) from 40.6 °C for extracts from untreated samples to 50.3 °C for the HPH-treated tomato peels suggests a more heterogeneous molecular structure in the cutin obtained through the HPH method compared to that extracted using conventional techniques [18]. The melting point of cutin extracted solely through conventional extraction processes from tomato peel has been reported in previous studies. Cifarelli et al. [18] investigated the thermal properties of cutin and observed multiple melting peaks ranging from 50 °C to 80 °C. In contrast, J. Benítez et al. [14] reported a single melting point at 55 °C for cutin extracted from tomato peel. These findings align to some extent with the range reported by our study.

### 3.5. Morphological Analysis

SEM analysis was utilized to visualize the surface morphology and structure of various samples, including tomato peels as received, those treated by conventional and HPH-assisted extraction processes, and the final cutin extracts obtained from both methods (Figure 6).

Untreated tomato peels exhibited a honeycomb structure with a uniform surface, devoid of any noticeable holes or cracks. Upon HSM treatment, the tomato tissue underwent fragmentation into smaller cell aggregates, with minimal impact on cell wall integrity, while retaining its honeycomb-like structure. Conversely, HPH treatment resulted in more pronounced individual cell wall breakage, disruption of cell structure, and the formation of cracks or pores with a honeycomb-like appearance, characterized by substantial cavities and high void segments. These findings are supported by the microscopic observations reported by Jurić et al. [24]. Their study revealed that while HSM treatment did not destroy individual cells of peel tissue, HPH treatment effectively disrupted these cells, particularly after 10 passes, resulting in complete disruption. Additionally, Xing et al. [30] explored the influence of HPH on the extraction of sulforaphane from broccoli seeds. Their SEM analysis corroborated cell rupture and cell wall breakage occurring during the HPH process. All the aforementioned alterations induced by HPH treatment were correlated with its strong mechanical effects on the microstructural surface of the tomato peels [24].

These findings indicate that the HPH technology proposed in this study facilitated a more significant degradation of tomato peel cells, enhancing the hydrolysis process and subsequently promoting cutin extraction from tomato peel.

Furthermore, the morphological analysis of cutin extracted from both untreated and HPH-treated tomato peels was also evaluated through SEM. The results revealed that cutin extracted from HPH-treated tomato peels exhibited a more irregular and heterogeneous structure compared to cutin extracted from untreated peels. This structural variance was likely attributed to the fluid-mechanical stresses applied during the HPH treatment or the coextraction of other compounds like cellulose during the cutin extraction process.

## 4. Conclusions

This study represents the first systematic investigation into the influence of key hydrolysis parameters (temperature, time, and NaOH concentration) and their interaction on cutin recovery from industrially derived tomato peels using a conventional extraction method. Response surface methodology (RSM) was instrumental in determining the optimal processing conditions (130 °C for 120 min, NaOH 3%) for maximizing cutin extraction yield.

Additionally, the potential of HPH pre-treatment of tomato peels prior to alkaline hydrolysis was explored to enhance cutin extraction while improving process sustainability. The results demonstrated that HPH technology increased cutin extraction yield by approximately 42% compared to the conventional process, while reducing solvent usage of NaOH and HCl by 33% and 35%, respectively. This enhancement was attributed to efficient tomato peel disintegration achieved through HPH pre-treatment, as confirmed by morphological analysis, streamlining the cutin extraction process and yielding higher output. FT-IR analysis verified the presence of characteristic functional groups in cutin samples from both conventional and HPH-assisted extraction processes from tomato peels. Moreover, thermogravimetric analysis (TGA) demonstrated thermal stability up to 200 °C, indicative of the inherent three-dimensional and heavily cross-linked cutin structure in samples extracted via conventional and HPH-assisted methods.

These results underscore the potential of HPH technology to intensify cutin recovery from tomato peels while allowing for gentler hydrolysis conditions.

## Figures and Tables

**Figure 1 foods-13-01415-f001:**
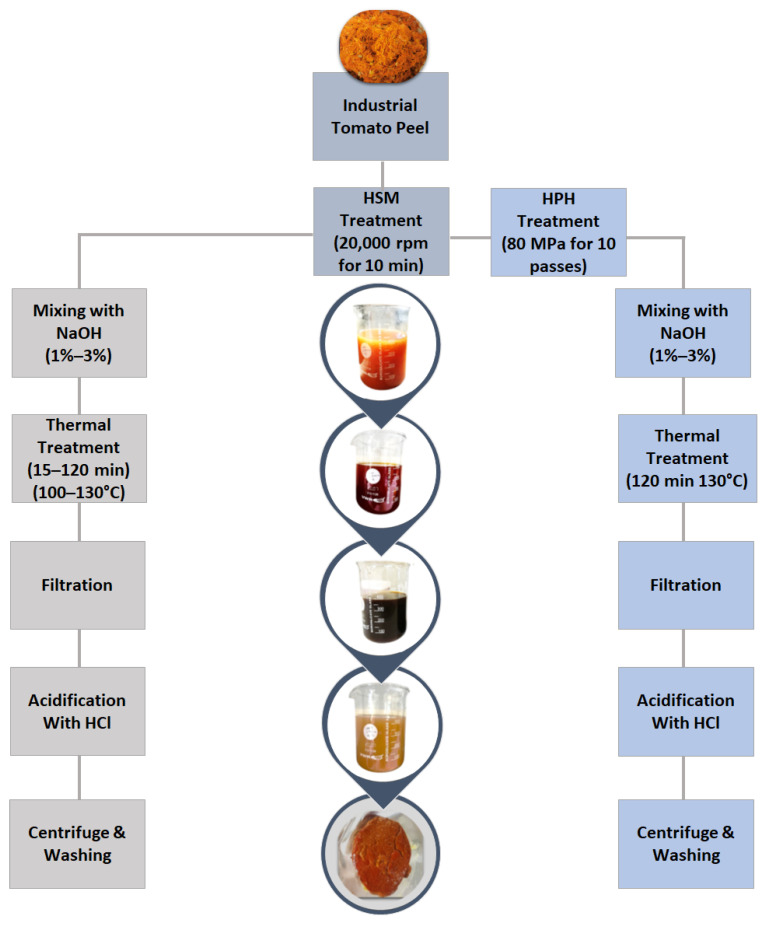
Schematic diagram of the procedure for extracting cutin from untreated (conventional process) and HPH-treated tomato peels. HSM: high shear mixing; HPH: high-pressure homogenization.

**Figure 2 foods-13-01415-f002:**
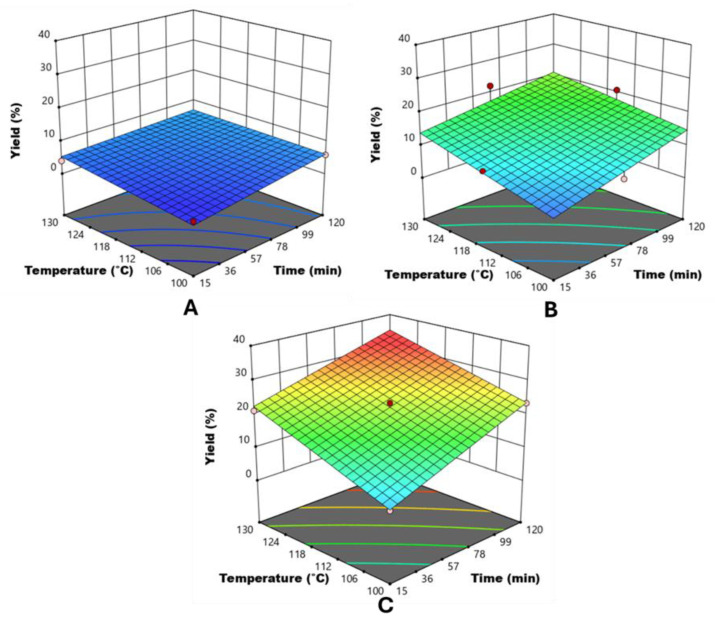
Response surfaces for the cutin yield achieved upon conventional extraction process from tomato peels as a function of hydrolysis time and temperature and for different NaOH concentrations: (**A**) 1% NaOH, (**B**) 2% NaOH, and (**C**) 3% NaOH.

**Figure 3 foods-13-01415-f003:**
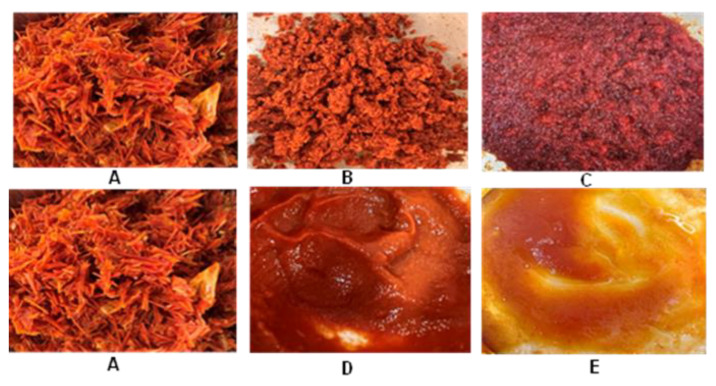
Images of untreated, HSM-treated, and HPH-treated tomato peels before and after alkaline hydrolyses (T = 130 °C, t = 120 min, 3% NaOH). (**A**) Untreated tomato peels, (**B**) HSM-treated tomato peels, (**C**) non-hydrolyzable residue after alkaline hydrolysis from HSM-treated tomato peels, (**D**) HSM-HPH-treated tomato peels, and (**E**) non-hydrolyzable residue after alkaline hydrolysis from HSM-HPH-treated tomato peels.

**Figure 4 foods-13-01415-f004:**
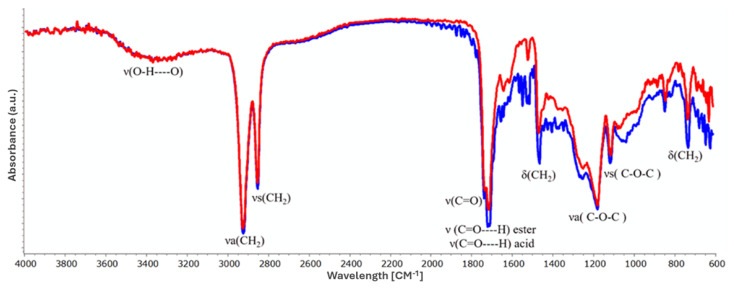
FT-IR absorbance spectra of cutin extracts obtained by conventional (blue line) and HPH-assisted (red line) extraction methods.

**Figure 5 foods-13-01415-f005:**
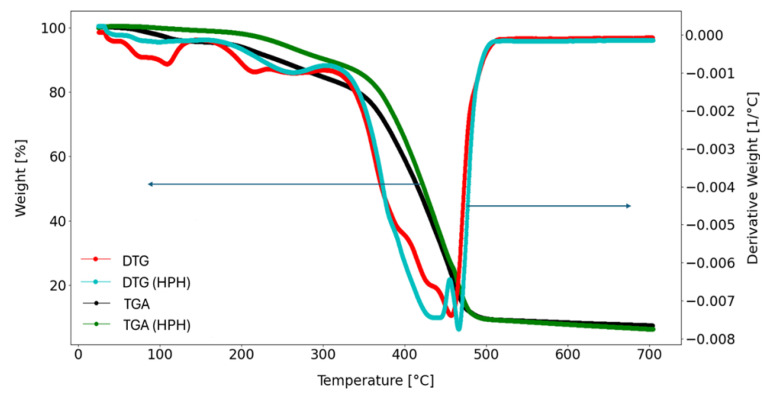
TGA and DTG graphs of cutin extracted from tomato peels in conventional (TP) and HPH-assisted (HPH-TP) extraction processes.

**Figure 6 foods-13-01415-f006:**
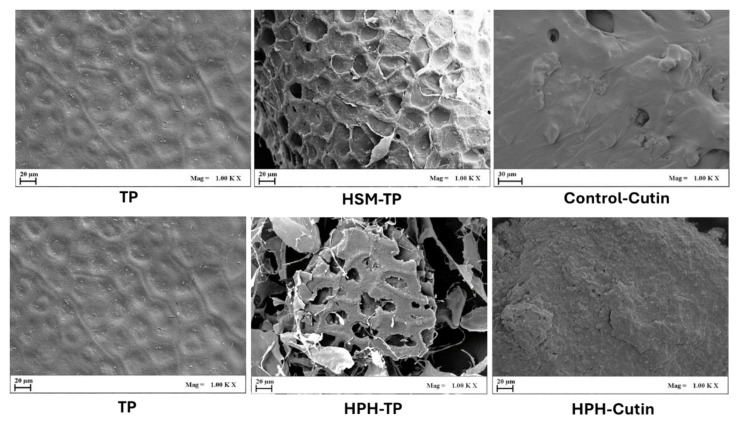
SEM images at 1000× magnification depicting tomato peel as received (TP), HSM-treated tomato peels (HSM-TP), HPH-treated tomato peels (HPH-TP), and cutin obtained through conventional extraction (control-cutin), and HPH-assisted extraction (HPH-cutin).

**Table 1 foods-13-01415-t001:** Effect of the independent factors investigated on the response variable (extraction yield) in cutin extraction from tomato peels through the conventional process.

Run	Variables			Response
	Time (min)	Temperature (°C)	NaOH (%)	Cutin Yield (%)
1	15	100	1	2.3 ± 0.74 _a_
2	15	100	3	7.3 ± 0.37 _b_
3	15	115	2	9.7 ± 0.57 _c_
4	15	130	1	3.9 ± 0.45 _d_
5	15	130	3	21.1 ± 0.40 _e_
6	67.5	100	2	7.4 ± 0.65 _b_
7	67.5	115	1	3.5 ± 0.64 _d_
8	67.5	115	3	23.4 ± 0.69 _f_
9	67.5	130	2	22.2 ± 0.90 _g_
10	120	100	1	5.8 ± 0.46 _h_
11	120	100	3	23.5 ± 0.82 _f_
12	120	115	2	20.9 ± 0.87 _e_
13	120	130	1	5.6 ± 0.37 _h_
14	120	130	3	32.5 ± 0.70 _i_

The results are expressed as mean ± standard deviation (*n* = 3 for factorial and axial points, *n* = 6 for central point). Values with different lowercase letters are significantly different (*p* ≤ 0.05).

**Table 2 foods-13-01415-t002:** Analysis of variance (ANOVA) of the second-order polynomial equation describing the influence of conventional extraction process parameters on the recovery yield of cutin.

Coefficients	Cutin Yield (%)	
β_0_	+1.08307	
β_1_ (t)	+0.096543	***
β_2_ (T)	−0.028729	**
β_3_ (NaOH)	−15.61802	***
β_12_ (t × T)	−0.001038	ns
β_13_ (t × NaOH)	+0.053190	*
β_23_ (T × NaOH)	+0.179667	*
*p*-value of the model	0.0002	***
R²	0.9578	
Adjusted R^2^	0.9217	
RMSE	0.00268	
*p* value of lack of fit test	0.47	ns

ns, not significant for *p* > 0.05; * significant for *p* ≤ 0.05; ** significant for *p* ≤ 0.01; *** significant for *p* ≤ 0.001; RMSE, Root Mean Square Error.

**Table 3 foods-13-01415-t003:** Yield of cutin from conventional extraction and HPH-assisted extraction from tomato peels at different NaOH concentrations.

Time (min)	Temperature (°C)	NaOH (%)	Yield (%)	
			Conventional Extraction	HPH-Assisted Extraction
120	130	1	5.6± 0.37_aA_	33.5± 0.78_aB_
120	130	2	25.3± 0.33_bA_	45.6± 0.45_bB_
120	130	3	32.5± 0.70_cA_	46.1± 0.86_bB_

Data are means ± standard deviations (*n* = 9). Values with different lowercase letters within the same column are significantly different (*p* ≤ 0.05). Values with different uppercase letters within the same row are significantly different (*p* ≤ 0.05).

**Table 4 foods-13-01415-t004:** Identification of the most relevant signals in the FT-IR spectra of cutin extracted by conventional and HPH-assisted extraction methods.

Wavelength (cm^−1^)		Chemical Bond
Conventional Extraction	HPH-Assisted Extraction	
3350	3355	ν(O-H----O)
2930	2930	νa (CH_2_)
2853	2851	νs (CH_2_)
1463 723	1463 724	bend (CH_2_)
1730	1730	ν (C=O) ester
1712	1713	ν (C=O----H) ester groups
1704	1705	ν (C=O----H) acid groups
1171	1170	νa (C-O-C) ester
1104	1103	νs (C-O-C) ester
1635	1632 (w)	ν (C=C) phenolic acid
1414	1417 (w)	ν(C-C) aromatic
1605 1456	1606 (w) -	ν(C-C) aromatic
1516 1554	1517 (w) 1552 (w)	ν(C-C) aromatic (conjugated with C=C) in phenolic compounds
834	835	Ɣ(C-H) aromatic

w: weak.

**Table 5 foods-13-01415-t005:** DSC analysis of cutin extracted through conventional (TP) and HPH-assisted (HPH-TP) extraction processes from tomato peels.

Sample Name	T_g_ (°C)	T_c_ (°C)	T_m_ (°C)
TP	−30.2	-	40.6
HPH-TP	−19.5	1.5	50.3

## Data Availability

The original contributions presented in the study are included in the article, further inquiries can be directed to the corresponding author.

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
