# Peer review of "Enhancing Cutin Extraction Efficiency from Industrially Derived Tomato Processing Residues by High-Pressure Homogenization"

_foods, 2024, doi:10.3390/foods13091415_

Round 1
Reviewer 1 Report
Comments and Suggestions for Authors
The aim of this study was to improve the recovery of cutin from tomato peels by HPH assisting the traditional extraction method, and to explain the effect of HPH on the extraction of cutin from tomato peels from the viewpoint of microstructure and compound type using SEM and FT-IR techniques.
There are several questions in the article:
1. In line 159, it is mentioned that "the solid-liquid ratio is maintained at 1:4 g/mL". How was this decision made? Is it supported by specific references?
2. The study used the RSM model to optimize the traditional extraction process, i.e., to investigate the effects of three variables, namely, time, NaOH concentration, and temperature, on the recovery of tomato peel cutin. How was the range of these three variables determined? A one-way test is recommended for additional data support.
3. The results of the RSM test showed a significant increase in tomato peel cutin extraction when the NaOH concentration was greater than 1%, which is a positive finding. However, the tomato peel cutin extraction rate did not show a significant turning point as the NaOH concentration increased. This raises the question: has the optimal extraction process been determined?
4. Although the RSM model was used in the study for the prediction of the optimal extraction process, proposing the use of a second-order polynomial model to predict the effects of the three independent factors on the response variable, the results section did not mention the conditions of the optimal extraction process as predicted by the model. What is the significance of the second order polynomial and ANOVA of the follow-up model? Do the optimal extraction conditions of RSM followed for the subsequent HPH-assisted extraction of cutin bias the experimental results?
5. In table 2, when using ANOVA to determine the significance of the RSM model, the model criterion should be set to p<0.05 and lack of fit >0.05. these are not mentioned in the text. please verify.
6. The text mentions that the RSM model has a correlation coefficient R2 of 0.9578, which is a very high correlation coefficient, close to 1. However, it is important to note that a high correlation coefficient does not fully represent an excellent fit of the regression model. Therefore, it is recommended to consider using the adjusted R2Adj to more accurately reflect the fit of the model.
7. The feature of this study should have been the effect of HPH-assisted extraction on cutin recovery. However, at the experimental design stage, only the interaction between HPH treatment and different concentrations of NaOH was considered, which made this part of the workload less. It is recommended that additional related experiments be conducted to explore this feature more fully.
Comments on the Quality of English Language
no
Author Response
Comments and Suggestions for Authors
The aim of this study was to improve the recovery of cutin from tomato peels by HPH assisting the traditional extraction method, and to explain the effect of HPH on the extraction of cutin from tomato peels from the viewpoint of microstructure and compound type using SEM and FT-IR techniques.
There are several questions in the article:
- In line 159, it is mentioned that "the solid-liquid ratio is maintained at 1:4 g/mL". How was this decision made? Is it supported by specific references?
The solid-liquid ratio was determined based on the previous work of Cifarelli et al. (2019). To ensure clarity, we have now included this reference in the text of section 2.3.
- The study used the RSM model to optimize the traditional extraction process, i.e., to investigate the effects of three variables, namely, time, NaOH concentration, and temperature, on the recovery of tomato peel cutin. How was the range of these three variables determined? A one-way test is recommended for additional data support.
According to this comment, now a sentence has been added to the text.
In particular, the ranges for the factors investigated in optimizing the conventional extraction process in this study were primarily derived from the patent of the conventional extraction procedure outlined by Cigognini et al. (2015). The suggested concentrations ranged from 0.5 M to 6 M (preferably 0.75 M), temperatures varied between 65°C and 130°C, and hydrolysis times spanned more than 15 minutes but less than 6 hours (preferably around 2 hours). Additionally, previous research indicated that temperatures lower than 100°C or very short of very long hydrolysis times were ineffective for cutin extraction. For instance, Cifarelli et al. (2016) demonstrated that the process yields obtained from hydrolysis reactions at 130°C for 2 hours were superior to those from treatments at 100°C for 6 hours or 130°C for 15 minutes. Consequently, the upper and lower limits of the temperature, time, and NaOH concentration domains were chosen based on the literature findings to cover a significant range of values.
As reported in the section of statistical analysis all the experiments and analyses were conducted in triplicate to calculate the mean and standard deviation (SD) for each set of experimental data. Differences with statistical significance between the means were determined using one-way ANOVA and Tukey’s Honest Significant.
- The results of the RSM test showed a significant increase in tomato peel cutin extraction when the NaOH concentration was greater than 1%, which is a positive finding. However, the tomato peel cutin extraction rate did not show a significant turning point as the NaOH concentration increased. This raises the question: has the optimal extraction process been determined?
The model fitting analysis revealed the presence of only linear and interaction terms, with the quadratic term showing non-significance. Consequently, we observed a rise in cutin yield as the treatment intensity increased, particularly with higher NaOH concentrations. Remarkably, the optimal condition identified through an internal procedure within the employed software coincided with the maximum processing conditions explored in this study, specifically 130°C, 2 hours, and 3% NaOH.
- Although the RSM model was used in the study for the prediction of the optimal extraction process, proposing the use of a second-order polynomial model to predict the effects of the three independent factors on the response variable, the results section did not mention the conditions of the optimal extraction process as predicted by the model. What is the significance of the second order polynomial and ANOVA of the follow-up model? Do the optimal extraction conditions of RSM followed for the subsequent HPH-assisted extraction of cutin bias the experimental results?
In the previous version of the manuscript, the optimal processing conditions identified through RSM analysis were indeed documented in Section 3.1. As explained in the text, the non-significance of the quadratic terms in the second-order model led to their exclusion from the overall model evaluation. Additionally, the ANOVA results, including RMSE, R-squared values, and the significance of the model based on p-values, were previously presented in Section 3.1. Moreover, in the updated version of the manuscript, the lack of fit has been incorporated into Table 2 and discussed.
Regarding the experiments involving HPH, as elaborated throughout the manuscript, the primary objective was to enhance cutin recovery while examining the interaction between the HPH process and NaOH concentration, which emerged as a more influential factor than time and temperature. Furthermore, the study aimed to investigate the potential of combining HPH technology with a gentler chemical hydrolysis process to develop an efficient and sustainable method for isolating cutin from tomato peels.
The inference drawn was that employing HPH pretreatment before the hydrolysis step under the optimal conditions determined via RSM significantly enhanced the recovery yield of cutin. Moreover, the utilization of HPH facilitated the reduction of NaOH concentration during the hydrolysis step conducted under the optimal thermal conditions (130°C, 120 minutes) determined via RSM. Overall, the results underscored the potential of HPH to intensify the cutin extraction process while operating under milder chemical hydrolysis conditions.
- In table 2, when using ANOVA to determine the significance of the RSM model, the model criterion should be set to p<0.05 and lack of fit >0.05. these are not mentioned in the text. please verify.
The model criterion was previously disclosed in the earlier version of the manuscript. Indeed, in the footnote of Table 2, it was clearly outlined as follows: "ns, not significant for p > 0.05. *Significant for p ≤ 0.05; **Significant for p ≤ 0.01; ***Significant for p ≤ 0.001. RMSE, Root Mean Square Error."
Now, following the Reviewer’s suggestion, the lack of fit has been added.
- The text mentions that the RSM model has a correlation coefficient R2 of 0.9578, which is a very high correlation coefficient, close to 1. However, it is important to note that a high correlation coefficient does not fully represent an excellent fit of the regression model. Therefore, it is recommended to consider using the adjusted R2Adj to more accurately reflect the fit of the model.
The adjusted R2 value has been now added to the Table 2. A value of 0.9217 was found that further confirm the goodness of the fit.
- The feature of this study should have been the effect of HPH-assisted extraction on cutin recovery. However, at the experimental design stage, only the interaction between HPH treatment and different concentrations of NaOH was considered, which made this part of the workload less. It is recommended that additional related experiments be conducted to explore this feature more fully.
We agree with the reviewer's suggestion that a comprehensive optimization of the HPH-assisted extraction process could offer valuable insights and enhance this innovative approach. However, as clearly stated in the main objective outlined at the end of the introduction section, this study aimed not only to optimize the conventional extraction process of cutin but also to assess whether implementing HPH pre-treatment of tomato peels could enhance the recovery yield of cutin during the conventional extraction process under the optimal thermal processing conditions determined through RSM analysis. Additionally, it aimed to evaluate the potential of HPH to achieve a specified cutin recovery under milder chemical hydrolysis conditions.
Nevertheless, we agree that future research endeavors should prioritize optimizing the processing conditions of this innovative extraction approach to fully leverage its potential. Therefore, we have included a sentence at the end of section 3.2, emphasizing the importance of conducting a systematic optimization of the innovative extraction method.
Reviewer 2 Report
Comments and Suggestions for Authors
L40-80: It can be reduced without losing the technical essence.
L115: Please discuss more on cutin extraction from fruits or vegetable sources.
L117-118: Please rewrite it.
L178-185: Not relevant.
L186: Why was FCCD selected instead of RCCD?
L188-190: Please justify the selection of the upper and lower limits of the domain.
L198: Why only second order model was used?
-Why do the author selected yield as the only response? How far is it realistic the one response optimization?
Figure 1: What hypotheses were behind employing HPH after the HSM step?
Section 2.5: The experimental design is not clear here.
L245-264: Please mention the reference and reduce the content.
L284-288: It looks it is more like a methodology statement, not from R&D.
L293: What about the degradation of cutin at that high temperature?
L350: Is the model developed coded for variables or real values of factors?
Table 2: Why were the square terms of any variables not included in the model? Were those square terms not significant?
L378-383: What about the lack of fit data?
L384-386: Please discuss more about the interaction effects like synergistic or antagonistic, or additive.
L393: This is only possible if those coefficients in Table 2 are in coded form.
L400-402: How was the optimization performed? Please provide the objective function expressions. Also, elaborate on the detailed methodology for optimization.
-The optimization point was 130C, 120 min, and 3% NaOH. This is the most sever or intense or extreme point of the domain. Therefore, how can it be justified? If you select to increase upper limits of variables, the optimization condition will vary accordingly. Please discuss this point in the revision.
L444-449: Same sentence repeated.
L470-472: Any further step to confrm this hypotheses?
L542-543: Which type of long chain compounds the authors are referring?
L609: Please connect it with the FTIR characterization.
L650: Please compare the data with similar literature reports.
L673: Please add a suitable reference.
Sec 3.5: Literature support and reasoning are missing.
L687-704: Please make it crisp. Report only the conclusion from this study. Please don’t extrapolate.
Author Response
The authors thanks the Reviewer for the careful reading of the manuscript and the useful comment that we are sure enabled to improve the manuscript. Enclosed below are detailed responses addressing each of your comments.
L40-80: It can be reduced without losing the technical essence.
The text has been streamlined to retain essential concepts, which we believe are crucial for aiding readers' comprehension of the results presented in Section 3.
L115: Please discuss more on cutin extraction from fruits or vegetable sources.
Further information (including references) about the use of innovative technologies of cutin extraction from fruits or vegetable sources have been added.
L117-118: Please rewrite it.
The sentence has been rewritten.
L178-185: Not relevant.
We concur with the Reviewer, and as such, the initial irrelevant portion of the paragraph has been eliminated.
L186: Why was FCCD selected instead of RCCD?
In experimental design, the choice between Full Factorial Central Composite Design (FCCD) and Rotatable Central Composite Design (RCCD) depends on various factors including the research objectives, the desired level of precision, and resource constraints.
However, FCCD typically offers greater flexibility in experimental design, allowing for the inclusion of additional factors or levels if needed without significant modifications to the design structure. If the experiment aims to investigate multiple factors and their interactions comprehensively, FCCD offers the advantage of assessing main effects, interactions, and quadratic effects simultaneously. In particular, it provides useful information about the linear and interaction effects of the factors, even though it may result less accurate in estimating pure quadratic coefficients. Furthermore, FCCD requires fewer experimental runs compared to RCCD to achieve the same level of information, making it more efficient in terms of time and resources.
Based on these considerations, we opted for FCCD over RCCD.
References:
- Szpisják-Gulyás, Aws N. Al-Tayawi, Zs. H. Horváth, Zs. László, Sz. Kertész3p and C. Hodúr (2023). Methods for experimental design, central composite design and the Box–Behnken design, to
optimise operational parameters: A review. Acta Alimentaria. DOI: 10.1556/066.2023.00235
Myers, R. H., Montgomery, D. C., & Anderson-Cook, C. M. (2016). Response surface methodology: process and product optimization using designed experiments (4th ed.). John Wiley & Sons.
L188-190: Please justify the selection of the upper and lower limits of the domain.
According to this comment, now a sentence has been added to the text.
In particular, the ranges for the factors investigated in optimizing the conventional extraction process in this study were primarily derived from the patent of the conventional extraction procedure outlined by Cigognini et al. (2015). The suggested concentrations ranged from 0.5 M to 6 M (preferably 0.75 M), temperatures varied between 65°C and 130°C, and hydrolysis times spanned more than 15 minutes but less than 6 hours (preferably around 2 hours). Additionally, previous research indicated that temperatures lower than 100°C or very short of very long hydrolysis times were ineffective for cutin extraction. For instance, Cifarelli et al. (2016) demonstrated that the process yields obtained from hydrolysis reactions at 130°C for 2 hours were superior to those from treatments at 100°C for 6 hours or 130°C for 15 minutes. Consequently, the upper and lower limits of the temperature, time, and NaOH concentration domains were chosen based on the literature findings to cover a significant range of values.
L198: Why only second order model was used?
This decision stemmed from the RSM results indicating that the model suggested by the software, as shown in the summary fit, was the most optimal.
-Why do the author selected yield as the only response? How far is it realistic the one response optimization?
The main objective of this study was twofold: firstly, to optimize the conventional extraction process of cutin, and secondly, to enhance the extractability of cutin while promoting sustainability through the integration of HPH into the conventional extraction process. We focused solely on cutin yield as the response variable and utilized various methodologies to characterize the quality of the extracts. These characterizations are crucial for assessing extract purity and potential applications. We did not identify any additional response variables of interest for our analysis. Consistent with our approach, previous studies in the literature have also investigated the effect of hydrolysis process parameters solely on cutin yield, even though without optimization (Cifarelli et al., 2019; Benítez et al., 2018). Therefore, we believe that focusing on only one response variable is realistic in this case.
Figure 1: What hypotheses were behind employing HPH after the HSM step?
The primary rationale for employing HPH after HSM, as detailed throughout the manuscript, is rooted in HPH's ability to achieve more intensive micronization of plant tissue with a uniform size distribution compared to HSM. This results in an increased surface-to-volume ratio and enhances the disruption of cellular structures, thereby likely creating favorable conditions for subsequent chemical hydrolysis and improving extraction efficiency, even when using low NaOH concentrations.
On the other hand, the use of HSM of plant tissues before HPH is necessary to sufficiently micronize the plant tissues, enabling them to pass smoothly through the orifice of the HPH system without causing any interruptions to the flow.
Section 2.5: The experimental design is not clear here.
The HPH-assisted extraction experiments did not utilize the RSM approach. As stated in the main objective outlined at the end of the introduction section, the RSM approach was solely applied to investigate the interaction between input factors and optimize the processing conditions of the conventional extraction process. Furthermore, at the beginning of section 2.5, it was explicitly mentioned that "The implementation of the HPH pre-treatment of tomato peels aimed to enhance the recovery yield of cutin during the conventional extraction process under the optimal thermal processing conditions previously determined through RSM analysis."
However, some little change to the text of section 2.5 was made to avoid any misunderstanding.
L245-264: Please mention the reference and reduce the content.
The content has been made more concise, preserving the key information, and a reference has been included.
L284-288: It looks it is more like a methodology statement, not from R&D.
We agree with the reviewer's comment, and the paragraph has been revised to better suit a section focused on results and discussion.
L293: What about the degradation of cutin at that high temperature?
As demonstrated in the thermal stability section (3.4) of this study and supported by previous research findings (Cifarelli et al., 2019; Benítez et al., 2018; Mellinas et al., 2022), the cutin extract exhibited an initial degradation temperature exceeding 200°C, which surpasses the highest temperature tested in this work. This confirms the excellent thermal stability of cutin.
L350: Is the model developed coded for variables or real values of factors?
The analysis was conducted using the actual, real values of the factors based on the results obtained from the 14 experimental runs reported in Table 1, without any transformation.
Table 2: Why were the square terms of any variables not included in the model? Were those square terms not significant?
According to the results of RSM analysis, the software Design Expert recommended employing a reduced second-order model. This model exhibited the best fit, as evidenced by the values of R-squared and RMSE, and it did not include square terms.
L378-383: What about the lack of fit data?
Now, in section 3.1 and Table 2, the lack of fit has been incorporated into the text. It was observed that the lack-of-fit value was 0.47, and the lack of fit test indicated non-significance (p > 0.05).
L384-386: Please discuss more about the interaction effects like synergistic or antagonistic, or additive.
From analysis of Variance (ANOVA) it emerged a significant positive interaction between NaOH and time and between NaOH and temperature, which provides evidence that the combined effect of the variables is indeed greater than what would be expected from their individual effects alone. Therefore, this supports the presence of a synergistic effect.
On the other hand, we observed a negative non significant interaction between T and time. In this case, while a negative interaction could hint at the potential for an antagonistic effect, it's essential to remember that without statistical significance, we can't confidently conclude that such an effect exists. It could simply be noise in the data or an indication that the sample size is too small to detect the true effect.
In our opinion, these are the only conclusion that can be derived from the ANOVA analysis, because the experimental set up do not include the effect of single input factors but only their combination.
Consequently, a statement emphasizing the presence of a synergistic effect between NaOH and time, as well as between NaOH and temperature, has been incorporated into section 3.1.
L393: This is only possible if those coefficients in Table 2 are in coded form.
The meaning of this comment is not entirely clear to us, particularly regarding the connection between the statement cited at L393 and the necessity to present the coefficient in coded form. However, as previously stated, we affirm that the results presented in Table 2 are derived from actual factor values, rather than coded representations.
L400-402: How was the optimization performed? Please provide the objective function expressions. Also, elaborate on the detailed methodology for optimization.
The optimal processing condition was determined using an internal procedure within the software employed. We chose not to include the objective function, as it can be readily obtained from Equation (2) by substituting the regression coefficients detailed in Table 2.
-The optimization point was 130C, 120 min, and 3% NaOH. This is the most sever or intense or extreme point of the domain. Therefore, how can it be justified? If you select to increase upper limits of variables, the optimization condition will vary accordingly. Please discuss this point in the revision.
The text has been modified accordingly, and a new sentence has been added to better explain this point.
L444-449: Same sentence repeated.
The repeated sentence has been deleted
L470-472: Any further step to confirm this hypotheses?
We agree that further exploration into the mechanism underlying HPH's ability to enhance cutin recovery is crucial. However, in this preliminary study, our primary focus was on observing the macroscopic effect of HPH on the extractability of cutin from tomato peels. Our SEM analysis confirmed the disruption of cell integrity in tomato peels following HPH treatment, leading to increased interaction between tomato peel particles and solvent, consequently enhancing cutin extraction compared to untreated controls. While we offered potential explanations for this mechanism, drawing from previous literature (Pirozzi et al., 2022), it's essential for future research to validate these hypotheses through further investigation.
L542-543: Which type of long chain compounds the authors are referring?
Generally, cutin consists of a combination of long-chain fatty acids, with two main types being C16 and C18. In the text, it has now been specified that we are referring to long-chain fatty acids.
L609: Please connect it with the FTIR characterization.
The text has been revised accordingly.
L650: Please compare the data with similar literature reports.
A comparison with findings of similar literature work has been added.
L673: Please add a suitable reference.
Unfortunately, as far as we know, there is no other work dealing with the cutin extraction aided by HPH technology.
Sec 3.5: Literature support and reasoning are missing.
Further literature has been included to support our findings.
L687-704: Please make it crisp. Report only the conclusion from this study. Please don’t extrapolate.
The text has been revised accordingly, and now includes only the main conclusion from the study.
Reviewer 3 Report
Comments and Suggestions for Authors
INTRODUCTION
1. Line 45, paragraph 1: .......Additionally, aromatic groups such as ......... Change to .......... Additionally, aromatic components such as dicarboxylic acids and glycerol........
2. Line 520
Figure 4. 3.3. FT-IR spectra to define the chemical structure of cutin
Line 533: ........ Furthermore, shoulders at 1730 and 1712 cm-1 were related to ester and carbonyls, ........
- I didn't understand! 1730 cm-1 would be stretching of C=O of normal ester, while 1712 cm-1, stretching of C=O of conjugated ester.
Change to: Furthermore, shoulders at 1730 and 1712 cm-1 were related to normal and conjugated ester carbonyls, respectively.
3. Lines 543 – 548: In the infrared spectrum, absorption bands around 1600 cm-1......at 835 cm-1.
Line 546: ..........with ν(C=C) stretching vibrations of phenolic acids at 1632 cm-1 , ν(C-C) aromatic at 1605 and 1606 cm-1 ........
- It's confusing: the bands around 1600 are not defined in the spectrum! ν(C-C) aromatic at 1605 and 1606 cm-1,...... : it is not aromatic.
- Band at 3350 cm-1 is too weak to define OH groups (water + phenols + acids...); Furthermore, OH group stretching band of carboxylic acids occur around 3000 cm-1 and not at 3350 cm-1
Suggestion: use the IR spectrum to characterize cutin, but limit its use: indicate group bands: C=O + C-C, + CH2 + C-O, etc. Reformulate Table 4, placing the data in a more objective way.
Author Response
INTRODUCTION
- Line 45, paragraph 1: .......Additionally, aromatic groups such as ......... Change to .......... Additionally, aromatic components such as dicarboxylic acids and glycerol........
This sentence has been changed accordingly.
- Line 520
Figure 4. 3.3. FT-IR spectra to define the chemical structure of cutin
We've opted to retain the figure caption from the previous manuscript version as we find it accurately describes the content of Figure 4.
Line 533: ........ Furthermore, shoulders at 1730 and 1712 cm-1 were related to ester and carbonyls, ........
- I didn't understand! 1730 cm-1 would be stretching of C=O of normal ester, while 1712 cm-1, stretching of C=O of conjugated ester.
Change to: Furthermore, shoulders at 1730 and 1712 cm-1 were related to normal and conjugated ester carbonyls, respectively.
The sentence was changed accordingly.
- Lines 543 – 548: In the infrared spectrum, absorption bands around 1600 cm-1......at 835 cm-1.
Line 546: ..........with ν(C=C) stretching vibrations of phenolic acids at 1632 cm-1 , ν(C-C) aromatic at 1605 and 1606 cm-1 ........
- It's confusing: the bands around 1600 are not defined in the spectrum! ν(C-C) aromatic at 1605 and 1606 cm-1,...... : it is not aromatic.
The bands have been checked and in few case revised. However, as far we know both spectra showed bands at 1605 cm−1 and at 835 cm−1 refer to the presence of aromatics, which is also corroborated by previous studies (Cifarelli et al., 2019; Mellinas et al., 2022).
- Band at 3350 cm-1 is too weak to define OH groups (water + phenols + acids...); Furthermore, OH group stretching band of carboxylic acids occur around 3000 cm-1 and not at 3350 cm-1.
We acknowledge the Reviewer's observation that the band at approximately 3350 cm-1 appears weak. This could be due to the fact that, generally, the broadness of the peak for OH group depends on the moisture content of the sample. In our samples the moisture of samples was relatively low and, likely also the observed band for OH was weak.
Nonetheless, our findings align with a previous study on cutin extraction from tomato peel, wherein the authors identified peaks around 3350 cm-1 as indicative of OH groups (Cifarelli et al., 2019; Mellinas et al., 2022; Heredia-Guerrero et al., 2014). For example, Cifarelli et al.(2019) cocnluded that “The broad, intense band at 3300 cm-1 was ascribed to the stretching of hydrogen-bonded hydroxyl groups ν(OH).”.
Suggestion: use the IR spectrum to characterize cutin, but limit its use: indicate group bands: C=O + C-C, + CH2 + C-O, etc. Reformulate Table 4, placing the data in a more objective way.
We thank the Reviewer for this suggestion. Following it, the text has been revised to eliminate any speculation, and Table 4 has been accordingly amended.